# Earlier menarche is associated with a higher prevalence of Herpes simplex type-2 (HSV-2) in young women in rural Malawi

Judith R Glynn[1]*, Ndoliwe Kayuni[2], Levie Gondwe[2], Alison J Price[1,2], Amelia C Crampin[1,2]

[1]Faculty of Epidemiology and Population Health, London School of Hygiene and Tropical Medicine, London, United Kingdom; [2]Karonga Prevention Study, Chilumba, Malawi

**Abstract** Remarkably little is known about associations between age at menarche and sexually transmitted infections, although girls with earlier menarche tend to have earlier sexual debut and school drop-out, so an association might be expected. In a population-based survey of >3000 women aged 15–30 in northern Malawi we show that those with earlier menarche had earlier sexual debut, earlier marriage and were more often Herpes simplex type-2 (HSV-2) positive. Compared to those with menarche aged <14, the age-adjusted odds ratios for HSV-2 were 0.89 (95%CI 0.71–1.1), 0.71 (0.57–0.89) and 0.69 (0.54–0.89) for menarche aged 14, 15 and 16+ respectively. This association persisted after adjusting for socio-economic factors, including schooling, and for sexual behaviour. No such association was seen with HIV infection, which is much less common and less uniformly distributed than HSV-2 in this population. The extra vulnerability of girls with earlier menarche needs to be recognised.

*For correspondence: judith.glynn@lshtm.ac.uk

Competing interests: The authors declare that no competing interests exist.

## Introduction

When a girl reaches menarche she may be regarded by some societies as 'ready' to start sex and to marry (**Sommer, 2009**). This may be linked to initiation ceremonies, recognising the transition to adulthood (**Munthali and Zulu, 2007**). Puberty may change her own desires, and change the behaviour of those around her. In diverse settings, earlier age at menarche has been associated with earlier sexual debut and earlier marriage (**Buga et al., 1996**; **Mensch et al., 2001**; **Downing and Bellis, 2009**; **Glynn et al., 2010**; **Boden et al., 2011**). Earlier menarche is also associated with school drop-out in some settings (**Biddlecom et al., 2008**; **Glynn et al., 2010**; **Boden et al., 2011**).

An association between early menarche and sexually transmitted infections might therefore be expected, but there is limited direct evidence to support this. In Croatia earlier menarche was associated with Chlamydial infection among sexually active adolescents attending outpatients (**Hirsl-Hecej et al., 2006**); in Tanzania earlier menarche was associated with human papilloma virus (**ter Meulen et al., 1992**); and in a prospective study in New Zealand earlier menarche was associated with attendance at an STI clinic before age 18 (**Boden et al., 2011**). A small study in Uganda found no link between age at menarche and HIV incidence. (**Quigley et al., 2000**) Since first coitus before menarche has been identified as a risk factor for sexually transmitted infections (**Duncan et al., 1994**), earlier menarche could also be protective in certain situations.

In many populations the prevalence of *Herpes simplex* type-2 (HSV-2), an important co-factor for HIV acquisition (**Glynn et al., 2009**), rises quickly after onset of sexual activity, so it provides a useful marker of unprotected sex, especially in women in whom the prevalence can reach high levels (**Obasi et al., 1999**).

**eLife digest** For many girls in sub-Saharan Africa their first menstrual cycle can mean an abrupt end to childhood because a girl's first period is often taken as a signal that she is 'ready' for sex and marriage. Most girls in this region have their first menstrual cycle between the ages of 13 and 18, with the median being around 15, and this is likely to get earlier as nutrition improves.

The age at which a girl in sub-Saharan Africa has her first period can have dramatic effects on her future prospects. Previous research in rural Malawi showed that more than half of those young women who had their first period before their 14th birthday never finish primary school and have sex before they are 16 years old. By comparison, 70% of women whose first menstrual cycle occurs at age 16 or older finish primary school and delay sex until after age 18.

To assess whether early menstruation might also be associated with the risk of contracting a sexually transmitted infection, Glynn et al. surveyed more than 3000 women between the ages of 15 and 30 in northern Malawi. Women who had their first menstrual cycle at an older age were less likely than those with an earlier onset of menstruation to be infected with Herpes simplex type-2. This difference persisted after adjustment for age at first sex and marriage, and for differences in socio-economic position, education and number of sexual partners. This may suggest that women with earlier onset of menstruation tended to have higher risk partnerships. There was no relationship between HIV and age at first menstrual cycle, probably because HIV is much less common than Herpes in the areas where the women lived.

Community and individual level interventions are needed to encourage and enable adolescent girls with early onset of menstruation to stay in school throughout puberty and beyond, and to help them reduce sexual risk taking from before their first sexual experiences.

In a study in rural northern Malawi we have previously shown that earlier menarche leads to earlier sexual debut, earlier marriage, and earlier school drop-out (*Glynn et al., 2010*). Here we investigate the associations with HSV-2 and HIV infection. In this area, unlike southern Malawi (*Morris, 2000*; *Munthali and Zulu, 2007*), there are no initiation rituals associated with menarche. Girls experiencing their first period are traditionally sent to stay with an aunt or other female relative for instruction, and it is likely to become known in the community.

## Results

Of 4772 women aged 15–30 who were eligible, 650 were not found and 146 refused to take part, leaving 3976, of whom 3965 were interviewed (83% of all those eligible). HIV results were available for 3651, and HSV-2 results for 3419. Those with missing HIV and HSV2 data were, on average, older, more educated, lived in better houses and experienced slightly later menarche.

Approximately one quarter of the women had menarche at each of <14, 14, 15 or ≥16 years of age. The associations between age at menarche, age at sexual debut and age at marriage are summarised in *Figure 1*. The age of sexual debut and the age at marriage increased with age at menarche. For example, among those with early menarche (age <14), 60% were married and only 16% were not yet sexually active at age 17, compared to 11% married and 71% not yet sexually active among those with late menarche (age ≥16).

Overall 25.5% of those tested were HSV-2 positive and 5.6% were HIV positive. The risk of both infections increased rapidly with age (*Figure 2*). After adjusting for current age, HSV-2 infection was less common in those with later menarche than in those with earlier menarche ($p$-trend = 0.001) but there was no evidence for a linear trend between age at menarche and HIV infection ($p$-trend = 0.8) (*Figure 3*; *Table 1*).

Some women who reported that they were not yet sexually active tested positive for HSV-2 and/or HIV (as previously noted) (*Glynn et al., 2011*). For both HSV-2 and HIV, the proportion positive increased with the lifetime number of partners, and was highest in those who had previously been married (divorced or widowed) and least common in those who had never married (*Table 1*).

HIV infection was more common near the main road; in those with more schooling, and whose parents had had more schooling; in those living in better-built houses; and in those working in the cash-economy. HSV-2 infection was equally common in the more distant areas as in the areas close to the road and was not associated with parental schooling or type of housing. It was least common among those with secondary

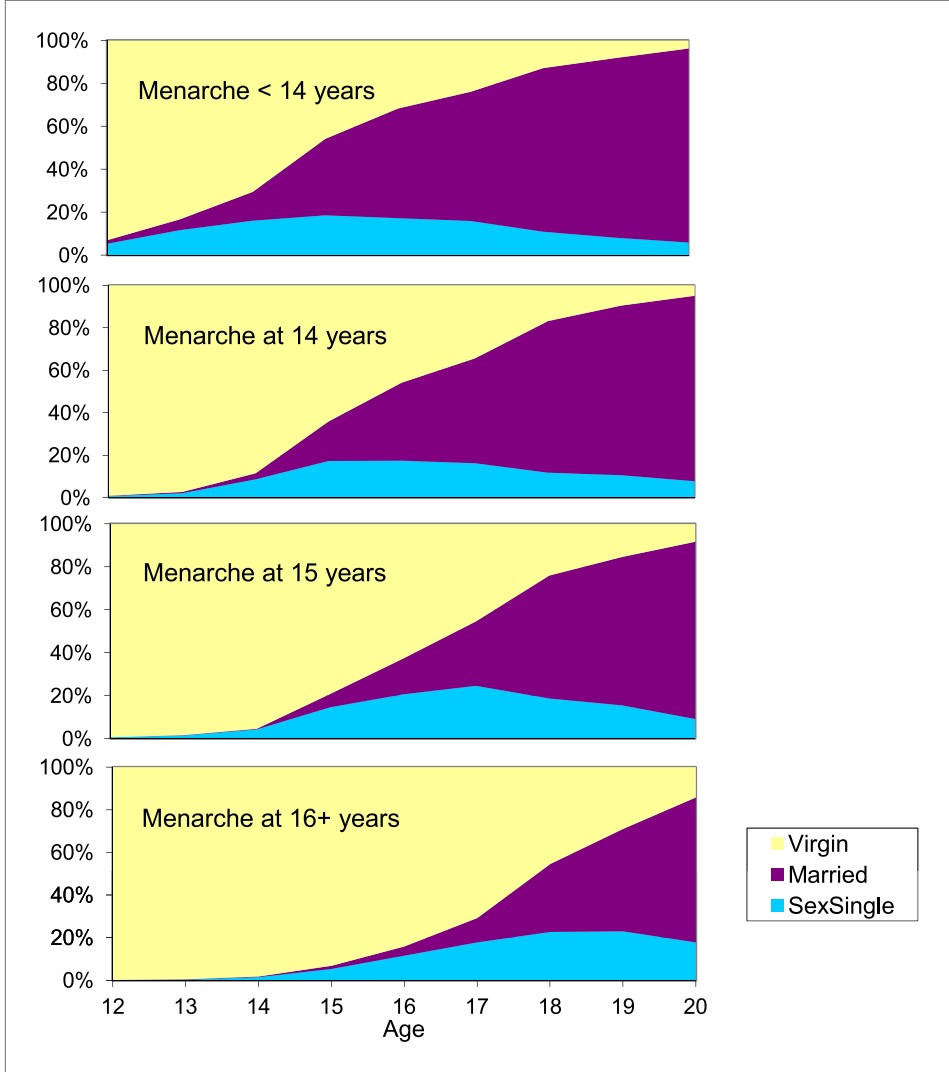

**Figure 1**. Multistate lifetables showing proportion of young women who are still virgins, have started sex but are unmarried, and are married, by age and age at menarche. Karonga District, Malawi.

or further education, and among those still in education. It was slightly less common among those who reported food insecurity. These trends persisted after adjusting for current age (*Table 1*).

*Table 2* shows the effect of additionally adjusting the association of age at menarche with HSV-2 or HIV for socio-economic variables and sexual behaviour variables. It is restricted to those with no missing data for the factors included to allow possible mediation by these factors to be fully explored. As shown in *Table 2*, adjustment for socio-economic and behavioural factors made no difference to the associations with HIV and only slightly weakened the trend with HSV-2. The factors included are those that were independently associated with HSV-2 or HIV infection in a full model, or that confounded the association between age at menarche and HSV2 or HIV. Additional adjustment for the other factors shown in *Table 1* made no further difference to the associations with HSV-2 or HIV.

## Discussion

Age at menarche in this population has a profound influence on adolescent trajectories through sexual debut and marriage. We have previously shown that early menarche is associated with school drop-out (*Glynn et al., 2010*). Here we show an association with HSV-2 infection, which is a marker of unprotected sex. The serological tests for HSV-2 have imperfect sensitivity and specificity (*van Dyck et al., 2004*), and since the resultant bias should be non-differential the true association with menarche could be stronger.

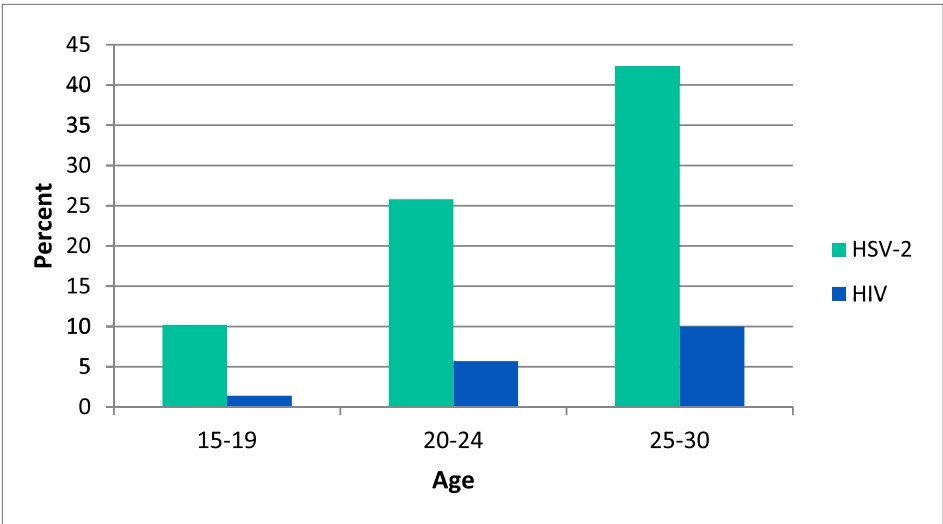

**Figure 2**. Prevalence of HSV-2 and HIV infection in women by age, Karonga District, Malawi.

In our study this association was not explained by measured socio-economic or behavioural variables. This suggests that it was influenced by unmeasured factors, such as the risk groups of the partners, or by factors that we have measured imprecisely. Limitations in reported sexual behaviour are well known, with a tendency for women to under-report onset of sexual activity and number of partners (*Buve et al., 2001*; *Glynn et al., 2011*; *Soler-Hampejsek et al., 2013*). The failure to explain the association may also be explained by the limitations of a cross-sectional study. The outcome was HSV-2 prevalence, not incidence, and the time of the infection was unknown. The age at menarche depended on recall. This would not be influenced by HSV-2 status, but could be influenced by (recalled) age at marriage or sexual debut.

If the association between early menarche and HSV-2 infection is due to riskier behaviour or higher risk partners, an association with HIV infection might also be expected. Especially as HSV-2 infection and HIV infection are strongly associated, in this population (not shown) and elsewhere. (*Glynn et al., 2009*) However, as shown in *Table 1*, HIV is much less uniformly distributed in the population and much less common, so the risk of a woman acquiring HIV depends less directly on her own or her partner's behaviour, and more on the local prevalence (and the probability that a given risk behaviour will lead to exposure to a partner with HIV infection), and on chance. The important role of population level rather than individual level risk factors for HIV has been shown in other settings (*Gabrysch et al., 2008*).

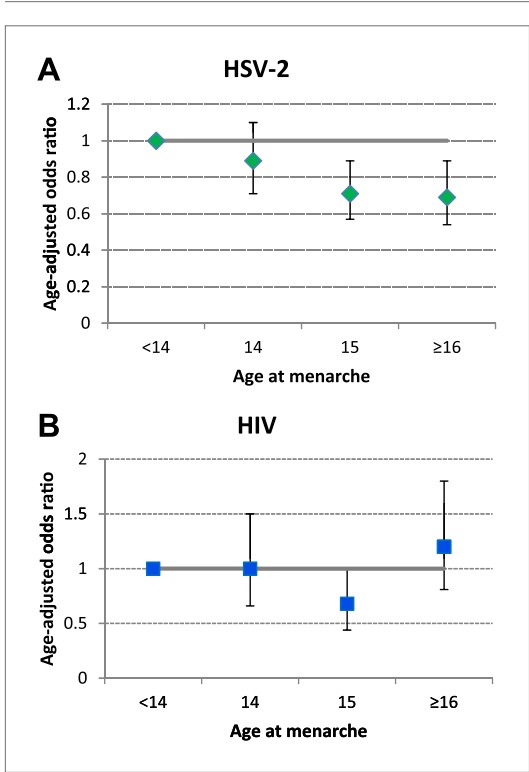

**Figure 3**. Association between age at menarche and (A) HSV-2 and (B) HIV, adjusted for current age. Karonga District, Malawi.

Early age at menarche appears to be a serious disadvantage to a girl's life chances in this and other populations (*Biddlecom et al., 2008*; *Downing and Bellis, 2009*; *Glynn et al., 2010*). As nutrition improves, age at puberty decreases (*Bellis et al., 2006*). Unless society's expectations

**Table 1.** Risk factors for HSV2 and HIV infection in 15–30 year old women in a population survey, Karonga District, Malawi 2007–8

| | HSV2 | | | Age adjusted | HIV | | | Age adjusted |
|---|---|---|---|---|---|---|---|---|
| | n/N | % | OR (95% CI) | OR (95% CI) | n/N | % | OR (95% CI) | OR (95% CI) |
| All | 870/3419 | 25.5 | | | 203/3651 | 5.6 | | |
| **Age** | | | | | | | | |
| 15–9 | 126/1240 | 10.2 | 1 | | 18/1314 | 1.4 | 1 | |
| 20–24 | 276/1072 | 25.8 | 3.1 (2.4–3.9) | | 64/1128 | 5.7 | 4.3 (2.6–7.4) | |
| 25–30 | 468/1107 | 42.3 | 6.5 (5.2–8.1) | | 121/1209 | 10.0 | 8.0 (4.8–13.2) | |
| **Age at menarche** | | | | | | | | |
| <14 | 245/815 | 30.1 | 1 | 1 | 51/861 | 5.9 | 1 | 1 |
| 14 | 195/769 | 25.4 | 0.79 (0.63–0.99) | 0.89 (0.71–1.1) | 43/819 | 5.3 | 0.88 (0.60–1.3) | 1.0 (0.66–1.5) |
| 15 | 225/960 | 23.4 | 0.71 (0.58–0.88) | 0.71 (0.57–0.89) | 41/1030 | 4.0 | 0.66 (0.43–1.0) | 0.68 (0.44–1.0) |
| ≥16 | 157/622 | 25.2 | 0.79 (0.62–0.99) | 0.69 (0.54–0.89) | 51/665 | 7.7 | 1.3 (0.88–2.0) | 1.2 (0.81–1.8) |
| **Age at first sex** | | | | | | | | |
| Never | 51/639 | 8.0 | 0.21 (0.15–0.29) | 0.48 (0.33–0.69) | 10/684 | 1.5 | 0.23 (0.12–0.44) | 0.69 (0.32–1.3) |
| <16 | 296/1014 | 29.2 | 1 | 1 | 66/1076 | 6.1 | 1 | 1 |
| 16–17 | 235/797 | 29.5 | 1.0 (0.83–1.2) | 0.99 (0.80–1.2) | 53/847 | 6.3 | 1.0 (0.70–1.4) | 0.99 (0.68–1.5) |
| 18+ | 273/930 | 29.4 | 1.0 (0.83–1.2) | 0.76 (0.62–0.93) | 71/1002 | 7.1 | 1.2 (0.82–1.7) | 0.90 (0.63–1.3) |
| **Marital status** | | | | | | | | |
| Never | 76/826 | 9.2 | 1 | 1 | 19/885 | 2.2 | 1 | 1 |
| Current | 681/2330 | 29.2 | 4.1 (3.2–5.2) | 1.7 (1.3–2.3) | 133/2481 | 5.4 | 2.6 (1.6–4.2) | 0.78 (0.44–1.4) |
| Previous | 113/263 | 43.0 | 7.4 (5.3–10.4) | 3.4 (2.4–5.0) | 51/285 | 17.9 | 9.9 (5.8–17.2) | 3.2 (1.7–5.9) |
| **Lifetime no. of partners** | | | | | | | | |
| 0 | 51/639 | 8.0 | 0.33 (0.24–0.46) | 0.76 (0.53–1.1) | 10/684 | 1.5 | 0.49 (0.25–0.98) | 1.4 (0.66–3.2) |
| 1 | 304/1475 | 20.6 | 1 | 1 | 46/1574 | 2.9 | 1 | 1 |
| 2 | 320/900 | 35.6 | 2.1 (1.8–2.6) | 2.0 (1.7–2.4) | 82/951 | 5.6 | 3.1 (2.2–4.5) | 2.9 (2.0–4.2) |
| 3 | 139/289 | 48.1 | 3.6 (2.7–4.6) | 3.2 (2.4–4.2) | 41/313 | 13.1 | 5.0 (3.2–7.8) | 4.4 (2.8–6.8) |
| ≥4 | 53/107 | 49.5 | 3.8 (2.5–5.6) | 3.5 (2.3–5.3) | 24/120 | 20.0 | 8.3 (4.9–14.2) | 7.4 (4.3–12.8) |
| **Distance from road** | | | | | | | | |
| <1 km | 410/1589 | 25.8 | 1 | 1 | 100/1595 | 6.3 | 1 | 1 |
| >1 km | 460/1830 | 25.1 | 0.97 (0.83–1.1) | 0.94 (0.80–1.1) | 62/1835 | 3.4 | 0.52 (0.38–0.72) | 0.50 (0.36–0.70) |
| **Schooling** | | | | | | | | |
| None/Primary 1–5 | 101/347 | 29.1 | 1 | | 16/378 | 4.2 | 1 | 1 |
| Primary 6–7 | 215/889 | 24.2 | 0.78 (0.59–1.0) | 0.88 (0.66–1.2) | 38/945 | 4.0 | 0.95 (0.52–1.7) | 1.1 (0.59–2.0) |
| Primary 8 | 271/1011 | 26.8 | 0.89 (0.68–1.2) | 0.89 (0.67–1.2) | 58/1069 | 5.4 | 1.3 (0.74–2.3) | 1.3 (0.75–2.4) |
| Secondary 1–3 | 218/860 | 25.4 | 0.83 (0.63–1.1) | 0.88 (0.65–1.2) | 70/916 | 7.6 | 1.9 (1.1–3.3) | 2.0 (1.2–3.6) |
| Secondary 4/Tertiary | 36/196 | 18.4 | 0.55 (0.36–0.84) | 0.42 (0.27–0.66) | 18/215 | 8.4 | 2.1 (1.0–4.1) | 1.8 (0.87–3.6) |
| **Mother schooling** | | | | | | | | |
| <=Primary | 716/2861 | 25.0 | 1 | 1 | 151/3021 | 5.0 | 1 | 1 |
| Secondary | 80/352 | 22.7 | 0.88 (0.68–1.1) | 1.0 (0.76–1.3) | 30/382 | 7.9 | 1.6 (1.1–2.4) | 1.8 (1.2–2.8) |
| **Father schooling** | | | | | | | | |
| <=Primary | 517/2046 | 25.3 | 1 | 1 | 94/2158 | 4.4 | 1 | 1 |
| Secondary | 260/1095 | 23.7 | 0.92 (0.78–1.1) | 0.96 (0.80–1.1) | 81/1166 | 7.0 | 1.6 (1.2–2.2) | 1.7 (1.3–2.3) |

*Table 1. Continued on next page*

*Table 1. Continued*

| | HSV2 | | | | HIV | | | |
| | | | | Age adjusted | | | | Age adjusted |
| | n/N | % | OR (95% CI) | OR (95% CI) | n/N | % | OR (95% CI) | OR (95% CI) |
|---|---|---|---|---|---|---|---|---|
| Housing quality | | | | | | | | |
| 1 (Best) | 148/662 | 22.4 | 1 | 1 | 71/732 | 9.7 | 1 | 1 |
| 2 | 149/565 | 26.4 | 1.2 (0.96–1.6) | 1.2 (0.89–1.5) | 32/598 | 5.4 | 0.53 (0.34–0.81) | 0.48 (0.31–0.75) |
| 3 | 317/1238 | 25.6 | 1.2 (0.96–1.5) | 1.0 (0.96–1.3) | 60/1296 | 4.6 | 0.45 (0.32–0.65) | 0.38 (0.26–0.55) |
| 4 (Worst) | 227/843 | 26.9 | 1.3 (1.0–1.6) | 1.2 (0.96–1.6) | 38/903 | 4.2 | 0.41 (0.27–0.61) | 0.37 (0.25–0.56) |
| Occupation | | | | | | | | |
| Farmer | 652/2229 | 29.3 | 1 | 1 | 143/2369 | 6.0 | 1 | 1 |
| In education | 63/720 | 8.8 | 0.23 (0.18–0.31) | 0.55 (0.40–0.76) | 10/767 | 1.3 | 0.21 (0.11–0.39) | 0.58 (0.28–1.2) |
| Not working | 23/77 | 29.9 | 1.0 (0.63–1.7) | 1.1 (0.66–1.9) | 8/88 | 9.1 | 1.6 (0.74–3.3) | 1.6 (0.77–3.5) |
| Other | 102/272 | 37.5 | 1.5 (1.1–1.9) | 1.3 (1.0–1.7) | 38/293 | 13.0 | 2.3 (1.6–3.4) | 2.1 (1.4–3.1) |
| Times of insufficient food in household in last year | | | | | | | | |
| No | 674/2575 | 26.2 | 1 | 1 | 162/2748 | 5.9 | 1 | 1 |
| Yes | 166/728 | 22.8 | 0.83 (0.69–1.0) | 0.83 (0.67–1.0) | 39/775 | 5.0 | 0.85 (0.59–1.2) | 0.85 (0.59–1.2) |
| Times when can't afford soap in last year | | | | | | | | |
| No | 519/2021 | 25.7 | 1 | 1 | 136/2166 | 6.3 | 1 | 1 |
| Yes | 321/1280 | 25.1 | 0.97 (0.82–1.1) | 0.97 (0.82–1.1) | 65/1355 | 4.8 | 0.75 (0.56–1.0) | 0.75 (0.55–1.0) |

OR = odds ratio

of a girl's role once she reaches puberty changes, increases in early sex, marriage and sexually transmitted infections can be expected. We have already discussed the need for community and individual level interventions to encourage and enable adolescent girls with early menarche to stay in school throughout puberty and beyond. To this should be added specific concerns over risky sexual behaviour in this group.

## Methods

Demographic surveillance was established in a population of approximately 33,000 individuals living in one area of Karonga District, northern Malawi, in 2002. The surveillance uses local key informants who register births and deaths continuously, and report them periodically to project staff. There was a re-census 2 years after registration was completed, and then annually thereafter. Full details are provided elsewhere. (*Jahn et al., 2007*; *Crampin et al., 2012*) In 2007, after extensive piloting, a sexual behaviour survey was added to follow the annual re-census, together with a serosurvey for HIV and HSV-2. Each sexual behaviour interview was conducted in the local language, in private. Written informed consent was requested separately for the interview and the serosurvey (*Glynn et al., 2011*). Ethics permission for the study was received from the Malawi Health Sciences Research Committee and the ethics committee of the London School of Hygiene & Tropical Medicine.

Data from the first sexual behaviour survey round were used for this analysis. A question on age at menarche for women was added midway through the first round. For those with missing data on age at menarche from the first round, the response from the subsequent round was used. Other questions included the participant's age at first sex, age at marriage, and number of partners, as well as schooling level and other socio-economic characteristics.

HSV-2 tests were conducted for women aged 15–30, using the type-2 specific enzyme immunoassay (Kalon Biological Ltd, Surrey, UK) which has the highest sensitivity and specificity of commercially available assays when used on African samples (*van Dyck et al., 2004*). HIV testing used rapid tests with the results immediately available to the participants (*Floyd et al., 2013*). We followed a parallel testing algorithm with Determine and Unigold, and SD-bioline as a tiebreaker if results were discordant, which we have previously shown to have very high sensitivity and specificity in this setting (*Molesworth et al., 2010*).

**Table 2.** Association between age at menarche and HSV2 and HIV, adjusted for other variables, among 15–30 year-old women in Karonga District Malawi

| Age at menarche | HSV2 (N = 3034)* | | | HIV (N = 3231)* | | |
|---|---|---|---|---|---|---|
| | Adjusted for age | Also adjusted for socio-economic variables | Also adjusted for behavioural variables | Adjusted for age | Also adjusted for socio-economic variables | Also adjusted for behavioural variables |
| | OR (95% CI) | OR (95% CI)† | OR (95% CI)‡ | OR (95% CI) | OR (95% CI)† | OR (95% CI)‡ |
| <14 | 1 | 1 | 1 | 1 | 1 | 1 |
| 14 | 0.91 (0.72–1.2) | 0.92 (0.73–1.2) | 0.92 (0.72–1.2) | 0.97 (0.63–1.5) | 0.94 (0.61–1.4) | 0.96 (0.62–1.5) |
| 15 | 0.73 (0.58–0.91) | 0.76 (0.60–0.95) | 0.77 (0.61–0.98) | 0.68 (0.44–1.0) | 0.64 (0.41–0.98) | 0.65 (0.42–1.0) |
| ≥16 | 0.70 (0.54–0.89) | 0.75 (0.58–0.97) | 0.77 (0.59–1.0) | 1.2 (0.79–1.8) | 1.1 (0.69–1.6) | 1.1 (0.70–1.7) |

*To allow the effects of these potential mediators to be explored fully, the analysis is restricted to individuals with no missing data for the factors included.

†age, schooling, occupation, lack of food.

‡age, schooling, occupation, lack of food, marital status, lifetime number of partners.

The association between age at menarche and ages at sexual debut and marriage were described using multistate life tables. The associations between age at menarche and other risk factors with HSV-2 and HIV were investigated using logistic regression. The extent to which the association between age at menarche and HSV-2 or HIV could be explained by measured behavioural and socio-economic factors was explored in multi-variable analyses, including these factors in the models, as appropriate. Analyses were done using STATA 12.1 (StataCorp, College Station, TX).

The data come from the Karonga Prevention study which has been conducting research in this population for more than 30 years. We have recently been awarded funding from the Wellcome Trust for a 4-year project to make our extensive data (on ~300,000 individuals and >1 million participant contacts) more accessible, and to ensure that, as we continue to add to this unique resource, it remains usable, flexible and available to local and international researchers. This involves making major changes to the data structure, the user interface and the way new data are collected, whilst maintaining the integrity and relationships in the existing database. Meanwhile, a procedure for data sharing is in place, which requires the consent of the programme director in consultation with senior science staff. No reasonable data sharing requests are turned down and the programme is committed to ensuring as much access as possible to the data, while maintaining full confidentiality. Enquiries to Dr Mia Crampin (mia.crampin@lshtm.ac.uk).

## Acknowledgements

We thank the Government of the Republic of Malawi for their interest in this Project and the National Health Sciences Research Committee of Malawi for permission to publish the paper.

## Additional information

### Funding

| Funder | Grant reference number | Author |
|---|---|---|
| Wellcome Trust | 079828/Z/06 | Judith R Glynn |

The funder had no role in study design, data collection and interpretation, or the decision to submit the work for publication.

### Author contributions

JRG, Conception and design, Analysis and interpretation of data, Drafting or revising the article; NK, LG, AJP, Acquisition of data, Drafting or revising the article; ACC, Conception and design, Acquisition of data, Drafting or revising the article

### Ethics

Human subjects: Written informed consent was requested separately for the interview and the serosurvey. Ethics permission for the study was received from the Malawi Health Sciences Research Committee (protocol number 419) and the ethics committee of the London School of Hygiene & Tropical Medicine.

## Additional files

### Major dataset

The following dataset was generated:

| Author(s) | Year | Dataset title | Dataset ID and/or URL | Database, license, and accessibility information |
|---|---|---|---|---|
| Crampin AC | 2013 | Karonga Prevention Study | | A procedure for data sharing is in place, which requires the consent of the programme director in consultation with senior science staff. No reasonable data sharing requests are turned down and the programme is committed to ensuring as much access as possible to the data, while maintaining full confidentiality. Enquiries to Dr Mia Crampin (mia.crampin@lshtm.ac.uk). |

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
