## [Decision Letter]

Thank you for sending your work entitled “Earlier menarche is associated with a higher prevalence of Herpes simplex type-2 (HSV-2) in young women in rural Malawi” for consideration at *eLife*. Your article has been favorably evaluated by Prabhat Jha as the Senior editor, Eduardo Franco as the Reviewing editor, and 2 peer reviewers, one of whom, Nico Nagelkerke, has agreed to reveal his identity.

The Reviewing editor and the reviewers discussed their comments before we reached this decision, and the Reviewing editor has assembled the following comments to help you prepare a revised submission.

1) Have you looked at the association between HSV-2 and HIV? Have you examined how replicable your findings are with respect to other STIs, i.e., is age at menarche negatively associated with risk?

2) Discuss potential recall biases for age at menarche. Is it possible that age at menarche is recalled in relation to onset of sexual behaviour? Other variables, such as absent fathers have been associated with early menarche. Was it possible to control for parental factors?

3) Was age at menarche based on a single question on first menstrual bleeding? Were there recurring questions that could have prevented measurement error? You mentioned social ceremonies associated with menarche in these populations. Was participation in them recorded?

4) Was the sample of respondents homogenous in terms of regions and cultures? There are many ethnic groups in Malawi and their social and nuptiality systems vary a lot. Could these differences explain the lack of association between HIV and menarche in the whole sample?

5) Are there HSV-2 differences between those with early menarche but married, and those with later menarche unmarried? In other words, is early marriage (following earlier menarche) protective? Could you entertain this point in a supplementary data analysis?

6) Please discuss biological hormonal changes for girls including biological sexually driven desire; you only mention “society's expectation of a girl's role”.

7) Consider the following comment from Dr. Nagelkerke intended to extract further insights from your data analysis: “Serological data are so-called ‘current status data’ and these are often analyzed *not* using logistic regression (i.e., with the ‘logistic link’), but with the complementary log-log link. This link allows interpretations in terms of the force-of-infection. As an ‘offset’ either the time since menarche or reported age of onset of sexual activity (or another choice made by the authors) could be used. Such an analysis does not need to replace their logistic regression analysis but could complement it.”

8) Please comment more clearly on the effect sizes with and without adjustment, so as to understand if these are as important as you claim.

9) We request that the tone of the discussion be more cautious, towards association and less to establish a causal link (given the relatively small size of the study, and it being in distinct populations).

---

## [Author Response]

*1) Have you looked at the association between HSV-2 and HIV? Have you examined how replicable your findings are with respect to other STIs, i.e., is age at menarche negatively associated with risk*?

There is a strong association between HSV2 and HIV, as expected, and this persists after adjusting for age and markers of sexual activity. We have not included this in the results of the paper as it is well known, but we have referred to it in the Discussion. We do not have data on any other STIs.

*2) Discuss potential recall biases for age at menarche. Is it possible that age at menarche is recalled in relation to onset of sexual behaviour? Other variables, such as absent fathers have been associated with early menarche. Was it possible to control for parental factors*?

The discussion already discusses recall biases and includes the following sentence:

“The age at menarche depended on recall. This would not be influenced by HSV-2 status, but could be influenced by (recalled) age at marriage or sexual debut.”

We do not have any data on absence of fathers. Note that we are not looking at “early” menarche, but “earlier”, with menarche in quartiles.

*3) Was age at menarche based on a single question on first menstrual bleeding? Were there recurring questions that could have prevented measurement error? You mentioned social ceremonies associated with menarche in these populations. Was participation in them recorded*?

There was a single two-part question: “Have you started your monthly period?” If “yes”, “How old were you when you started your monthly period?” (It is asked in the local language.)

In this area, unlike southern Malawi, there are no initiation ceremonies. We have clarified this in the Introduction.

*4) Was the sample of respondents homogenous in terms of regions and cultures? There are many ethnic groups in Malawi and their social and nuptiality systems vary a lot. Could these differences explain the lack of association between HIV and menarche in the whole sample*?

All respondents come from one small area. We have clarified this in the Methods.

*5) Are there HSV-2 differences between those with early menarche but married, and those with later menarche unmarried? In other words, is early marriage (following earlier menarche) protective? Could you entertain this point in a supplementary data analysis*?

There is a limit to which this can be explored with a cross-sectional study, not least because most women in this population are married by age 20. We have looked at the subgroup of those aged 17–19. In this group those with menarche <14 who were married had a similar prevalence of HSV-2 as unmarried women with menarche at 16+ (∼12% in each group). We have not added this to the paper.

*6) Please discuss biological hormonal changes for girls including biological sexually driven desire; you only mention “society's expectation of a girl's role”*.

We say in the Introduction “Puberty may change her own desires”.

*7) Consider the following comment from Dr. Nagelkerke intended to extract further insights from your data analysis: “Serological data are so-called ‘current status data’ and these are often analyzed* not *using logistic regression (i.e., with the ‘logistic link’), but with the complementary log-log link. This link allows interpretations in terms of the force-of-infection. As an ‘offset’ either the time since menarche or reported age of onset of sexual activity (or another choice made by the authors) could be used. Such an analysis does not need to replace their logistic regression analysis but could complement it.*”

For this analysis we have concentrated on establishing that there is an association between age at menarche and HSV-2, and that this is not explained by age or measured sexual behaviour factors. For this purpose we do not feel that the further analysis, which would estimate rate ratios and rates rather than odds ratios, would add anything to the interpretation. Furthermore, it would be difficult to define a suitable offset. Individuals would be at risk from sexual debut but we know some individuals who claimed never to have had sex were infected, so they would be excluded if we used that.

*8) Please comment more clearly on the effect sizes with and without adjustment, so as to understand if these are as important as you claim*.

We have reworded this in the Results.

*9) We request that the tone of the discussion be more cautious, towards association and less to establish a causal link (given the relatively small size of the study, and it being in distinct populations)*.

We have altered the wording.